# Geolocation Assessment and Optimization for OMPS Nadir Mapper: Methodology

**Likun Wang** [1,*]**, Chunhui Pan** [1]**, Banghua Yan** [2]**, Trevor Beck** [2]**, Junye Chen** [3]**, Lihang Zhou** [4]**, Satya Kalluri** [4] **and Mitch Goldberg** [4]

1. Cooperative Institute for Satellite Earth System Studies (CISESS), Earth System Science Interdisciplinary Center (ESSIC), University of Maryland, College Park, MD 20742, USA; chunhui.pan@noaa.gov
2. NOAA/NESDIS/Center for Satellite Applications and Research, College Park, MD 20740, USA; banghua.yan@noaa.gov (B.Y.); trevor.beck@noaa.gov (T.B.)
3. Global Science and Technology, Inc., Greenbelt, MD 20770, USA; junye.chen@noaa.gov
4. NOAA/NESDIS/JPSS Program Office, Greenbelt, MD 20771, USA; lihang.zhou@noaa.gov (L.Z.); satya.kalluri@noaa.gov (S.K.); mitch.goldberg@noaa.gov (M.G.)
* Correspondence: wlikun@umd.edu; Tel.: +1-301-683-3551

**Abstract:** Onboard both the Suomi National Polar-orbiting Partnership and Joint Polar Satellite System (JPSS) series of satellites, the Ozone Mapping and Profiler Suite Nadir Mapper (OMPS-NM) is a new generation of a total ozone column sensor and is used to generate total column ozone products. This study presents a method for efficiently assessing OMPS-NM geolocation accuracy using spatially collocated radiance measurements from the Visible Infrared Imaging Radiometer Suite (VIIRS) Moderate Band M1 by taking advantage of its high spatial resolution (750 m at nadir) and accurate geolocation. The basic idea is to find the best collocation position with maximum correlation between VIIRS collocated and real OMPS-NM radiances by perturbing OMPS-NM line-of-sight (LOS) vectors in the cross-track and along-track directions with small steps in the spacecraft coordinate. The perturbation angles at the best collocation position where OMPS-NM and VIIRS are optimally aligned are used to characterize OMPS-NM geolocation accuracy. In addition, the assessment results can be used to optimize the OMPS-NM field view angle lookup table in the Sensor Data Record (SDR) processing software to improve its geolocation accuracy. To demonstrate the methodology, the proposed method is successfully employed to evaluate OMPS-NM geolocation accuracy with different spatial resolutions. The results indicate that, after the view angle table was updated, the geolocation accuracy for both SNPP and NOAA-20 OMPS-NM is on the sub-pixel level (less than ¼ pixel size) along all the scan positions in both cross-track and along-track directions and the performance is very stable with time. The method proposed in this study lays down the framework for assessing the geolocation accuracy of future high-resolution OMPS-NM measurements.

**Keywords:** geolocation; calibration; Ozone Mapping and Profiler Suite; Visible Infrared Imaging Radiometer Suite; image registration

## 1. Introduction

The Ozone Mapping and Profiler Suite (OMPS) is a new generation of space-based sensor suite that provides measurements of global three-dimensional distribution of atmospheric ozone and other constituents [1–3]. OMPS is a suite of instruments that consists of three spectrometers: (1) a nadir mapper (OMPS-NM) that maps global ozone, (2) a nadir profiler (OMPS-NP) that measures the vertical distribution of ozone in the stratosphere [4,5], and (3) a limb profiler (OMPS-LP) that provides ozone in the lower stratosphere and troposphere with a high vertical resolution. The OMPS instruments have been carried out on the Suomi NPP (SNPP) (launched on 28 October 2011) and NOAA-20 (N20) satellites (launched on 18 November 2017) and will be on a future Joint Polar Satellite System (JPSS)

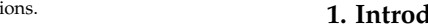



series of satellites from JPSS2 through JPSS4. Except for the NOAA-20, which only carries NM and NP, all three instruments are on the SNPP and will be on other JPSS satellites.

Among these three instruments, the OMPS-NM is a total ozone column sensor and uses a single grating and a charge-coupled device (CCD) to make measurements every 0.42 nm from 300 to 380 nm with 1.0 nm full width at half maximum (FWHM) resolution. The OMPS-NM Sensor Data Records (SDRs) are used to generate the total column ozone products daily, including total column ozone, effective reflectivity, and ultraviolet-absorbing aerosols (an aerosol index) [1]. Therefore, the data quality of the OMPS-NM SDR directly impacts its downstream environmental products. Intensive postlaunch calibration efforts have been carried out by the NOAA OMPS SDR team, which focuses on postlaunch spectral, radiometric and geometric calibration, and algorithm improvements [6–10]. Geometric calibration is used to accurately map OMPS-NM line-of-sight (LOS) vectors from the OMPS-NM macropixels to their geodetic latitude and longitude on the Earth ellipsoid surface. This is one of the important factors that impacts OMPS-NM SDR data quality. More importantly, the geolocation parameters are directly used as inputs to calculate the solar zenith angles for a given OMPS observation location and thus can impact downstream ozone retrievals. Thus, it is always critical to ensure that the SDR data meet geolocation accuracy specifications. During the final review of the SDR data (an official JPSS program review to approve the data quality level), the OMPS-NM geolocation accuracies at nadir for the two OMPS NMs were within the requirement (less than 5 km at nadir) [9]. Recently, it was found that geolocation errors can be around 80.0 km at far-off nadir pixels for both SNPP and NOAA-20 OMPS NM SDR data by comparing OMPS-NM with VIIRS data on the Earth Surface [11]. Efforts were conducted to derive new Field Angle Map (FAM) lookup tables (LUTs) to reduce off-nadir pixel geolocation errors for the two OMPS NMs. The new LUTs for the two OMPS-NMs have been implemented in the NOAA Interface Data Processing Segment (IDPS) OMPS operational processing stream since 28 June 2021. This effort highlights the significance of establishing an effective method that can efficiently detect the geolocation problem, particularly for off-nadir pixels at cross-track and along-track positions. This is especially important for upcoming JPSS-2 and future OMPS-NM since a similar off-nadir geolocation error problem should exist. In addition, it is important to produce new FAM LUTs based on quantitative geolocation assessment results for future OMPS NM sensors.

Following the above efforts to improve OMPS NM geolocation accuracy, a method of using the Visible Infrared Imaging Radiometer Suite (VIIRS) high-resolution observations to quantitatively assess OMPS-NM geolocation accuracy is developed in this study. The key is to simulate coarse-resolution OMPS-NM images from high-resolution VIIRS data based on OMPS-NM geometry characteristics. Pixel-to-pixel registration is then performed between the VIIRS-collocated and real OMPS-NM images to characterize the OMPS-NM geolocation errors by assuming VIIRS geolocation fields as a reference. Because both OMPS and VIIRS are onboard the same satellite platform, the proposed method can be applied to any operational OMPS-NM SDR data at any location. Finally, the proposed method can perform postlaunch on-orbit geometric calibration by optimizing the OMPS-NM FAM LUTs based on the assessment results. A similar method has been successfully applied to the Cross-track Infrared Sounder (CrIS) geolocation assessment and optimization using VIIRS [12,13]. The root causes of geolocation errors will be presented in the following study [14].

The paper is organized as follows: Section 2 summarizes the instrument and data, Section 3 discusses the nature of geolocation assessment, Section 4 describes the method, Section 5 provides the application of the methods, and Section 6 concludes the paper.

## 2. Instruments and Datasets

### 2.1. OMPS-NM

OMPS-NM is a total ozone column sensor that measures the radiance spectra from 300–380 nm every 0.42 nm per FWHM [3]. Specifically, the telescope with a dichroic filter redirects the photons from the Earth View into the OMPS-NM spectrometer, which has

a 110° total across-track field of view, resulting in 2800 km instantaneous coverage at the Earth's surface. The light from the NM spectrometer is dispersed via diffraction grating onto a two-dimensional CCD located at the spectrometer's focal plane. Along the spectral dimension, the CCD consists of 340 pixels, covering a spectral range of 300 to 380 nm. In the cross-track spatial dimension, there is a total of 740 pixels and most of the pixels are used in operation. During nominal operation, the CCD pixel signals along the cross-track direction are aggregated and form separate "macropixel" fields of view (FOVs) through a sample table. Furthermore, the individual pixels on an Earth-view CCD are integrated by exposure time in the along-track spatial direction to form the required horizontal cell sizes. In other words, OMPS nadir temporal (along-track) and spatial (across-track) resolutions are highly configurable because their macropixels are constructed in programmable flight software.

Table 1 shows the three datasets with different spatial configurations used in this study. The NOAA operational OMPS-NM SDR products—including the OMPS Nadir Total Column Science SDR (SOMTC) and OMPS Total Column Ellipsoid Geolocation (GOTCO)—are the focus of this study. With a spatial resolution of 50 km (along-track) by 50 km (cross-track) at nadir for SNPP and 17 km by 50 km for NOAA-20, these data are processed by the IDPS from the Raw Data Records (RDR) and are obtained from NOAA's Comprehensive Large Array-data Stewardship System (CLASS). To clearly demonstrate the methodology, we use the OMPS-NM data on NOAA-20 as an example, while the assessment results for SNPP OMPS-NM are also presented. For the future OMPS-NM on JPSS2, the spatial resolution will be further improved. Therefore, the medium-resolution data on NOAA-20—which were generated from NASA's Science Investigator-led Processing System (SIPS)—is used as a proxy to test the proposed method, which has a spatial resolution of 17 km (along-track) by 12.5 km (cross-track) at nadir.

**Table 1.** OMPS-NM SDR datasets with different spatial configurations.

| Satellite Platform | Cross-Track | | Along-Track | | Producer |
|---|---|---|---|---|---|
| | Nadir Resolution (km) | Macropixels per Scan | Integration Time (Seconds) | Nadir Resolution (km) | |
| SNPP | 50 | 35 | 7.5 | 50 | NOAA IDPS |
| NOAA-20 | 50 | 35 | 2.5 | 17 | NOAA IDPS |
| NOAA-20 | 12.5 | 140 | 2.5 | 17 | NASA SIPS |

### 2.2. VIIRS

A detailed description of the VIIRS instrument can be found in [15]. With 22 spectral bands covering the spectrum between 0.412 μm and 12.01 μm, the VIIRS instrument is a whiskbroom scanning radiometer with a field of regard of ±56.3° in the cross-track direction with a swath width of about 3060 km. In this study, the VIIRS Moderate Resolution Band 1 SDR (SVM01) with a central wavelength of 412 nm and Moderate Bands SDR Ellipsoid Geolocation (GMODO) with a 750 m resolution (at nadir) is used because this band is close to the OMPS-NM spectra and both OMPS-NM and VIIRS can detect similar spectral features when they are sensing the same scenes. The VIIRS geolocation accuracy has been assessed through the correlation between the Ground Control Point data sets from Landsat [16,17]. The mean errors are about 26.0 and 13.0 m, while the root-mean-square errors are about 78.0 m and 60.0 m in the in-track and cross-track directions, respectively. Compared to the OMPS-NM coarse resolution, the VIIRS geolocation dataset is sufficiently accurate and thus can be treated as a truth in this study.

Figure 1a,b shows an example of the VIIRS band M1 (412 nm) and correspondent NOAA20 OMPS-NM data (NOAA SDR) at 380 nm observed at 07:07 UTC on 29 May 2021. Both instruments observed the Earth from the same satellite platform simultaneously. As a high-spatial resolution imaging system, the VIIRS can clearly capture the fine structures of deep connective clouds, while these structures look very fuzzy on low-resolution OMPS-NM images. In addition, its image resolution gradually becomes coarser from the nadir to the off-nadir views due to the macropixel FOV size expansion. However, because of the

different spectral responses of VIIRS and OMPS-NM (412 nm versus 380 nm), the VIIRS cloudy radiance values are apparently larger than OMPS-NM.

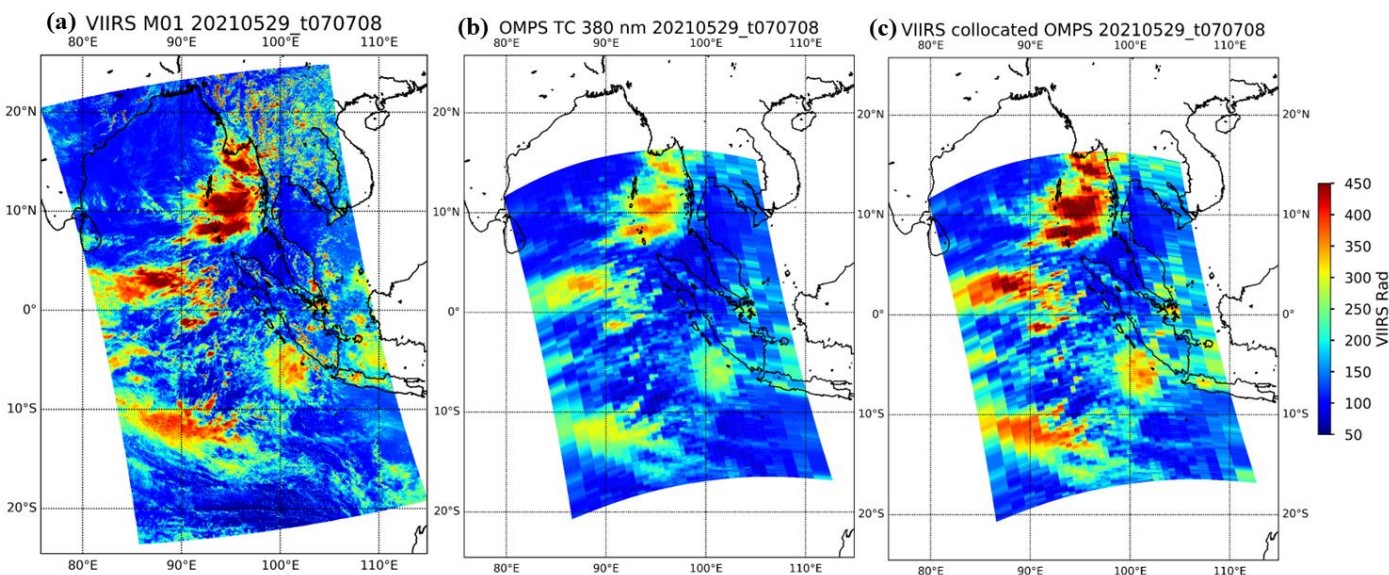

**Figure 1.** (**a**) VIIRS image at the M01 band (412 nm) on 29 May 2021; (**b**) the correspondent OMPS-NM image at 380 nm; (**c**) VIIRS-collocated image based on the OMPS-NM geometry characteristics. The radiance unit is W/(cm$^3$ Sr) in the images.

## 3. Nature of OMPS-NM Geolocation Assessment

### 3.1. OMPS-NM Geolocation Algorithm Overview

The OMPS-NM geolocation algorithm maps OMPS-NM LOS pointing vectors to geodetic longitude and latitude on the Earth's surface at each macropixel. Specifically, its geolocation calculation is divided into two parts, i.e., the sensor-specific algorithm and spacecraft level one. Figure 2 is a flow chart that illustrates the procedures for OMPS-NM geolocation calculations; the black arrows indicate the procedures for the geolocation calculations [16]. A detailed description of these coordinate systems can be found in [18,19] or in Table 1 in [13].

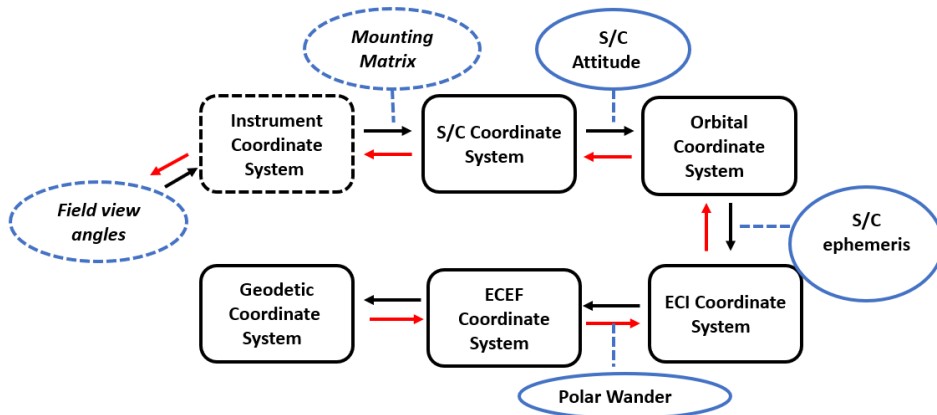

**Figure 2.** Flowchart that illustrates the procedures for OMPS-NM geolocation calculation, in which the black and red arrows indicate the forward and inverse geolocation calculations, respectively. The blocks with the dashed line represent the sensor-specific algorithm, while the blocks with the solid lines stand for the spacecraft-level algorithm. The blue blocks are the data used for the coordinate transformation, where the field angles and mounting matrix are static, but others are dynamic (varying with time).

In the sensor-specific geolocation algorithm, theoretically, the view vectors for each of the CCD pixels are computed in instrument coordinates with the help of optical ray trace models. Transformation matrices can be developed for each of the optical subassemblies between the focal plane and entrance aperture and then combined to calculate the view vector. For operational use, the OMPS sensor-specific geolocation algorithm begins with a set of view angles for CCD pixels to simplify optical ray trace computations. The optical view angles ($\alpha$, $\beta$) provided are expressed in the form of rotational angles, and the unit exit vectors ($x$, $y$, $z$) in the instrument coordinate are calculated as:

$$x = \sin(\beta)$$
$$y = -\sin(\alpha) * \cos(\beta)$$
$$z = \cos(\alpha)\cos(\beta), \tag{1}$$

where $\alpha$ is called by as the azimuth look angle (rotation around the $x$-axis), and $\beta$ as the elevation look angle (rotation around the $y$-axis). The alignment between the instrument and spacecraft coordinates were measured during the spacecraft integration and is expressed as the instrument mounting matrix ($3 \times 3$) in the sensor-specific geolocation algorithm. After these two steps, the view vectors for the CCD pixels are expressed in the spacecraft coordinates. In summary, only two sets of parameters are involved in the sensor-level geolocation computations, i.e., the optical view angle lookup table and the instrument mounting matrix, both of which are static.

Shared by all instruments, the spacecraft-level algorithm computes geodetic longitude and latitude at given universal time coordinated (UTC) time by finding the intersection of the LOS vectors with the Earth ellipsoid [19,20]. The following steps are involved:

(1) Using the spacecraft attitude to transform the LOS vector from the spacecraft coordinate to the orbital coordinate;
(2) Transforming the LOS vector from the orbital coordinate to the Earth-centered inertial (ECI) coordinate systems based on the spacecraft's ephemeris data (ECI position and velocity);
(3) Converting the LOS vector from ECI to the Earth-centered Earth-fixed (ECEF) coordinate;
(4) Outputting geodetic latitude and longitude through the Earth ellipsoid intersection algorithm; and
(5) Computing other geolocation products, such as solar and sensor azimuth and zenith angles, as well as the sensor range (from satellite to macropixel FOV position).

*3.2. Geolocation Uncertainty*

The OMPS-NM geolocation uncertainties are mostly from sensor-level mapping knowledge uncertainty because all the instruments on the SNPP and JPSS satellites share the common spacecraft-level algorithm and use the common inputs from the spacecraft diary files. The parameters (e.g., satellite ephemeris, platform attitude and Earth polar motion data) in the common geolocation algorithm are dynamic (or vary with time) and shared by all the instruments. Based on the VIIRS geolocation accuracy assessment results [16], the uncertainties from the common geolocation computation should be negligible compared to the OMPS-NM geolocation specification and spatial resolution.

At the instrument level, the contributors to the mapping knowledge uncertainty of OMPS-NM can be divided into two main components: dynamic and static. The static components include prelaunch measurements in alignment at the instrument level and, additionally, for any shift that occurs during spacecraft installation and launch. The geolocation error from static components is usually shown as a systematic error. The dynamic components include uncertainties related to instrument drift and jitter for both in-track and cross-track directions and uncertainties attributed to thermal distortions. The dynamic contribution is the smaller component of mapping uncertainty and varies with time and location. This study mainly aims to identify the mapping knowledge uncertainty of OMPS-NM from the static components, i.e., the optical view angle lookup table and

instrument mounting matrix. Note that the $3 \times 3$ instrument mounting matrix is used to align the instrument coordinate with the S/C coordinate, but its transformation impacts are very small (on the order of $10^{-3}$ degree).

## 4. Methods for Geolocation Accuracy Assessment

### 4.1. Collocation between OMPS-NM and VIIRS

The fast and accurate collocation of OMPS-NM with VIIRS observations is vital in this study. Onboard the same satellite platform, VIIRS and OMPS-NM observe the Earth from the same satellite platform almost simultaneously, with different instrument view characteristics and spectral responses. The collocation between OMPS-NM and VIIRS is to accurately collect high-spatial resolution VIIRS pixels falling within the relatively large OMPS-NM macropixel FOV footprint. Therefore, as the first step, one needs the OMPS-NM macropixel projected FOV shape, which is saved as a variable in the OMPS-NM geolocation dataset. Figure 3a shows an example plot of the OMPS-NM macropixel FOV shape (blue polygon) projected on the Earth surface around each FOV center (red dots). As a result, the collocation process is used to extract the underlying VIIRS pixels within each FOV. A G-ring or G-polygon (defined as a closed, connected and contiguous area by the anchor great circle arcs) technique [21,22], has been developed to extract pixels from geometry features (circle, polygons or shape files) on the Earth ellipsoid surface (i.e., the LLA coordinate system in Table 1 in [13]). An alternative way is to perform the collocation in the S/C coordinate, in which the OMPS-NM macropixel FOV shapes are almost identical (as shown in Figure 3b). This approach requires an efficient and precise inverse geolocation algorithm (described below) that can quickly transform VIIRS geolocation fields and OMPS macropixel FOV shapes from the LLA coordinate into the S/C coordinate through vectorized computation.

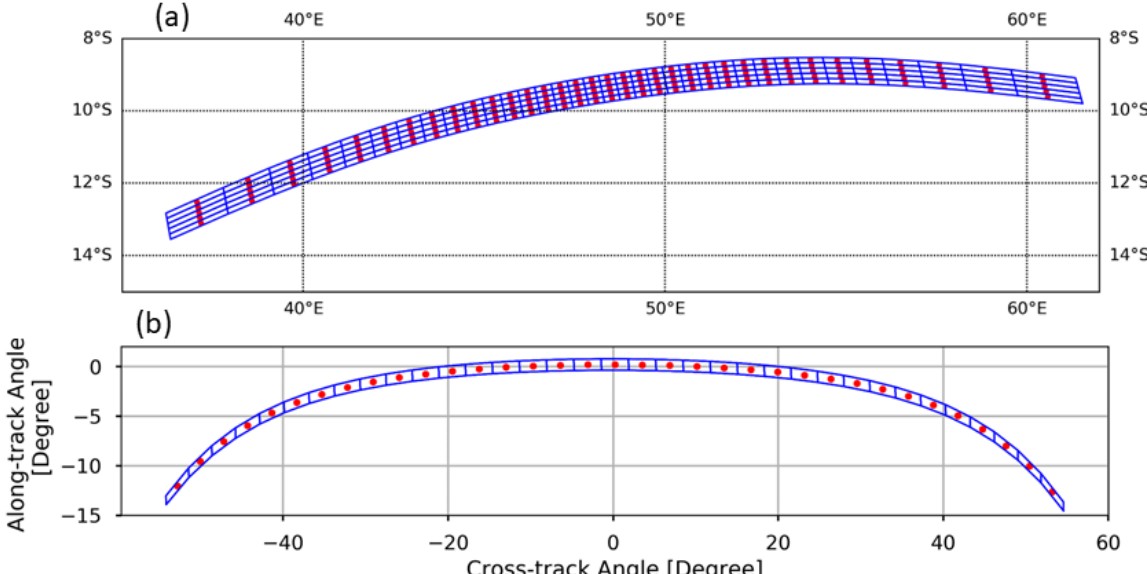

**Figure 3.** Example of OMPS-NM macropixel FOV shapes projected (**a**) on the Earth Surface, shown as longitude and latitude and (**b**) on the S/C coordinate expressed in the along-track and in-track angles (indicated by the bluelines). The red dots point to the FOV center. The FOVs on the Earth's location in five scanlines return to identical locations on the S/C coordinate after the inverse geolocation computation.

Using the OMPS-NM view geometry characteristics, the VIIRS pixels are collocated within each OMPS-NM macropixel and then averaged together to simulate the low-resolution OMPS-NM image, which is shown in Figure 1c and hereafter is called the VIIRS-collocated OMPS-NM observations. After collocation, both real and VIIRS-collocated OMPS-NM images reveal similar deep convective cloud structures with the same spatial resolution and view geometry, while the radiation is from different spectral responses

(412 nm vs. 380 nm). Ideally, if OMPS-NM perfectly aligns with VIIRS in space, their radiance difference image should disclose only the spectral difference of the target structures (e.g., clouds). However, the radiance difference image (Figure 4a) clearly reveals random noise patterns at off-scan positions (mixed warm and cold bias), indicating that the VIIRS and OMPS-NM are not perfectly aligned together. This is clearly demonstrated in Figure 5—a scatter plot between the real and VIIRS-collocated OMPS-NM radiances. The large spread (or standard deviations) is mostly caused by the misalignment between VIIRS and OMPS. The aim of this study is to quantify this misalignment between VIIRS and OMPS-NM, i.e., OMPS-NM geolocation errors. Therefore, one needs to transform the OMPS-NM images (source) to align with the VIIRS-collocated OMPS-NM image (target) through a perturbation process. The amount of perturbation needed in each macropixel indicates the geolocation errors of OMPS-NM relative to the VIIRS geolocation fields.

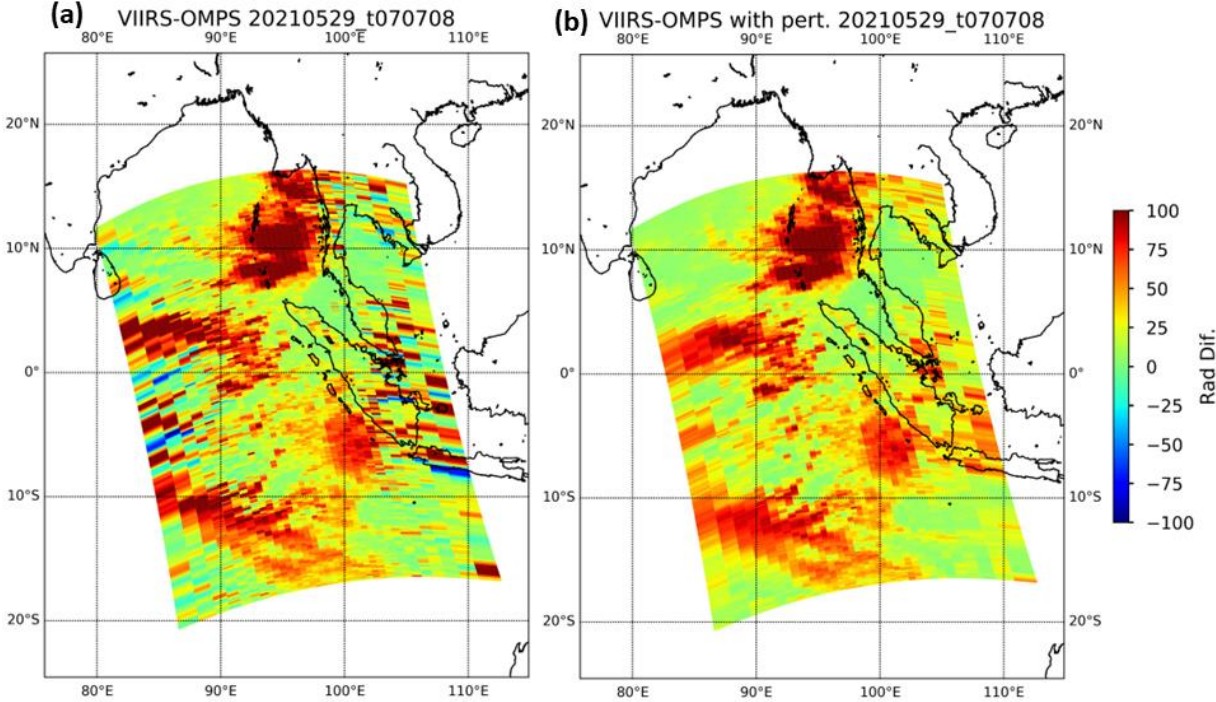

**Figure 4.** Radiance difference image between real and VIIRS-collocated OMPS-NM radiances: (**a**) based on the original OMPS geolocation fields and (**b**) the ones with perturbation that perfectly are aligned with VIIRS. The radiance unit is $W/(cm^3\ Sr)$.

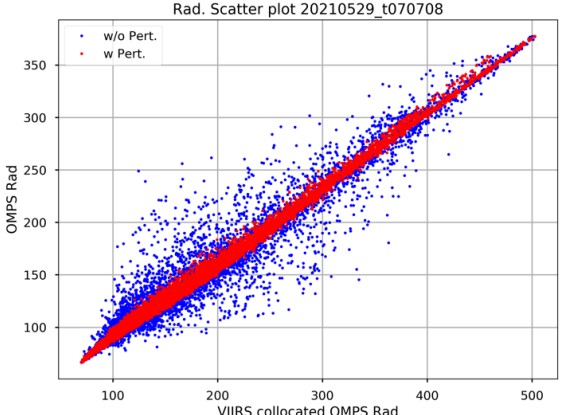

**Figure 5.** Scatter plot between real and VIIRS-collocated OMPS-NM radiances. The blue dots indicate the VIIRS-collocated OMPS-NM radiances using the original OMPS geolocation fields, while the red dots are perfectly aligned with the VIIRS. The radiance unit is $W/(cm^3\ Sr)$.

### 4.2. Inverse Geolocation Computation

As discussed above, to identify the geolocation errors caused by the sensor-level mapping parameters, one needs to perturb the LOS vectors of OMPS-NM macropixels to find the best set that is well aligned with VIIRS. This perturbation bypasses the coordinates used in the spacecraft-level algorithm and avoids any dynamic or time-varying parameters. Therefore, we chose to perform perturbation in the S/C coordinate. To do that, inverse geolocation computation is needed to derive the OMPS-NM LOS vectors in the S/C coordinate, which are indicated by the red arrows in Figure 2. The details on each transformation have been described in [13] and these steps are not repeated. After inverse geolocation calculations, the OMPS geolocation data of each pixel expressed originally in longitude and latitude on the Erath surface are transformed back as a unit vector in the S/C coordinate, as shown in Figure 6. Therefore, the along-track (*x-z* plane) and cross-track planes (*y-z* plane) in the S/C coordinate can be defined. Once the unit OMPS vector LOS (*x*, *y*, *z*) is projected onto these two planes, two angles of θ and φ—which indicate a relative location of the OMPS LOS vector LOS relative to the along-track and in-track planes—can be simultaneously computed as follows:

$$\theta = arctan\left(\frac{x}{z}\right)$$

$$\varphi = arctan\left(\frac{y}{z}\right) \qquad (2)$$

Relevantly, this transformation changes a three degrees of freedom unit vector [*x*, *y*, *z*] into two angles of θ and φ, thus making the perturbation of the LOS vectors easy and simple.

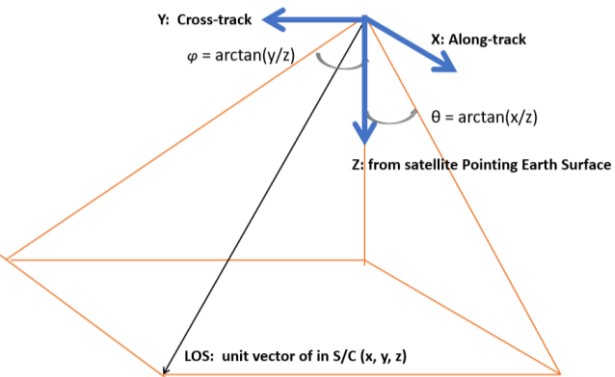

**Figure 6.** The OMPS LOS vector is expressed in the spacecraft coordinate, where the along-track (*x-z* plane) and cross-track planes (*y-z* plane) are formed. Two angles of θ and φ simultaneously correspond to the unit vector of **LOS**.

Shown in Figure 3b is an example of the OMPS-NM macropixel FOV spatial distribution in terms of their along-track and cross-track angles (θ, φ) in the S/C coordinate through the inverse geolocation computation, which is originally from their Earth surface locations shown in Figure 3a (including five scans). In theory, if the inverse geolocation is accurate and precise enough, the retrieved along-track and cross-track angles should only be related to the FOV index, even though they are from different scanlines on Earth's surface. This is because the forward geolocation calculation begins from the same set of static field view angles, which are only dependent on the CCD index. The inverse calculation should also end up with a similar FOV index-dependent relationship. To examine this, we computed the along-track and cross-track angles in the S/C coordinate from the granule shown in Figure 1a, which includes a total of 150 scanlines. The mean and standard deviation of these angles are given in Figure 7. The standard deviation along all the macropixel FOV positions is on the level of $10^{-6}$ degree, indicating that the inverse geolocation computation is accurate with negligible uncertainties.

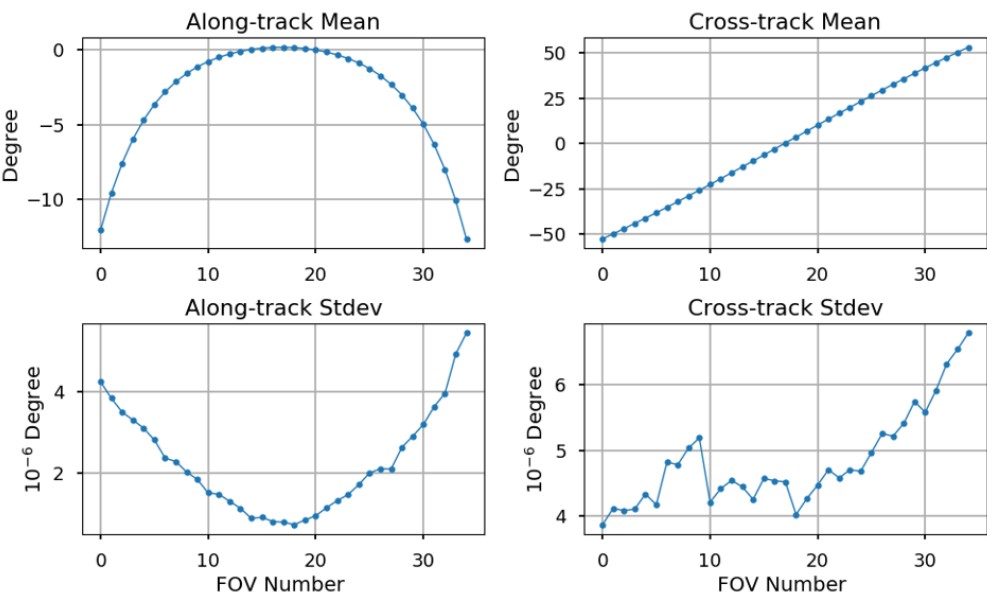

**Figure 7.** The along-track (**left**) and cross-track (**right**) angles in the S/C coordinate computed for the granule shown in Figure 1a, which contains 150 scanlines. The mean and standard deviations are given at the top and bottom, respectively.

### 4.3. Perturbation of OMPS-NM LOS Vectors

After the inverse geolocation computation, the OMPS geolocation data expressed originally in longitude and latitude on the Erath Surface are transformed as unit vectors in the spacecraft coordinate, which are further characterized using the along-track and cross-track angles ($\theta$, $\varphi$). One can perturb the LOS unit vector by changing $\theta$ and $\varphi$ angles as follows:

$$\theta_i^{fov} = \theta_0^{fov} + i \times \Delta s + \theta_{guess}^{fov}, \quad i = -\frac{m-1}{2}, -\frac{m-2}{2} \dots, 0, \dots \frac{m-2}{2}, \frac{m-1}{2}$$

$$\varphi_j^{fov} = \varphi_0^{fov} + j \times \Delta s + \varphi_{guess}^{fov}, \quad j = -\frac{n-1}{2}, -\frac{n-2}{2} \dots, 0, \dots \frac{n-2}{2}, \frac{n-1}{2}, \quad (3)$$

where $\theta_0^{fov}$ and $\theta_0^{fov}$ are the along-track and cross-track angles derived from the OMPS-NM geolocation dataset and only dependent on the macropixel FOV index, $\theta_{guess}^{fov}$ and $\varphi_{guess}^{fov}$ is the first guess value and can be estimated based on the initial assessment results, and $\Delta s$ is the perturbation step angle, and m and n the total perturbation steps in the along-track and cross-track direction, respectively. An example of the perturbation angle setup is given in Figure 8. Specifically, the blue dots are the along-track and cross-track angles derived from the OMPS-NM geolocation dataset corresponding to $\theta_0^{fov}$ and $\theta_0^{fov}$ in Equation (3). The red dots are the first guess angles that are $\theta_{guess}^{fov}$ and $\varphi_{guess}^{fov}$ in Equation (3). After an initial assessment, we empirically set up the first guess angle to guarantee that the perturbation range can cover the potential geolocation errors. The red bar indicates the perturbation range. In the along-track steps, the perturbation is carried out from $-1.5°$ to $1.5°$ with m steps, while it was performed from $-1.3°$ to $1.3°$ with n steps. The step angle is set as 0.1 degrees. m and n are arbitrary numbers and can be chosen based on geolocation errors. We chose 31 and 27 for this study.

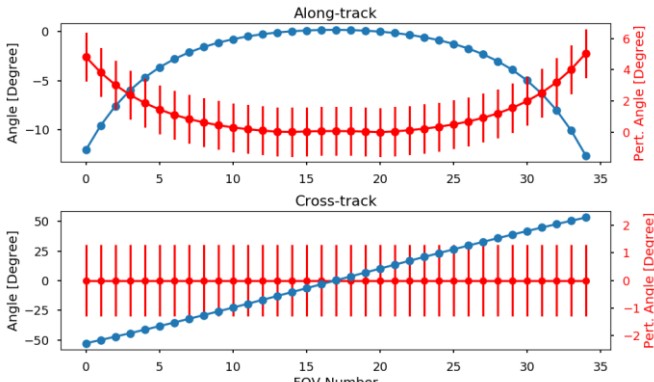

**Figure 8.** Example of the perturbation angle set up. The blue dots are the along-track and cross-track angles derived from the OMPS-NM geolocation dataset, which correspond to $\theta_0^{fov}$ and $\theta_0^{fov}$ in Equation (3). The red dots are the first guess angles that are $\theta_{guess}^{fov}$ and $\varphi_{guess}^{fov}$ in Equation (3). The red bar indicates the perturbation range. In the along-track steps, the perturbation begins from $-1.5$ to 1.5 degrees with a step angle of 0.1 degrees (31 steps), while it is performed from $-1.3$ to 1.3 degrees with a step angle of 0.1 degrees (27 steps).

### 4.4. Determination of OMPS-NM Geolocation Accuracy

Following Equation (4), a total of m $\times$ n (31 $\times$ 27) unit vectors $\overrightarrow{LOS}_{i,j}^{FOV}$ in the S/C coordinate are generated, which can be expressed in terms of $(x, y, z)$ as follows:

$$\overrightarrow{LOS}_{i,j}^{FOV}(x,\ y,\ z) = Unit\left(z * \tan\left(\theta_i^{fov}\right),\ z * \tan\left(\varphi_j^{FOV}\right),\ z\right) \tag{4}$$

So, the key question become that, among a total of m $\times$ n (31 $\times$ 27) unit vectors, which one is best aligned with VIIRS.

For each OMPS-NM $\overrightarrow{LOS}_{i,j}^{FOV}$ vector, one can generate a new set of geolocation datasets through forward geolocation computation. The high spatial resolution VIIRS image can be collocated with each new OMPS-NM geolocation dataset to generate its related VIIRS-collocated OMPS radiances (Figure 1c). As a result, the correlation coefficients of the original OMPS radiance data with a total of m $\times$ n (31 $\times$ 27) sets of VIIRS-collocated OMPS-NM radiances can be calculated separated with each macropixel FOV position. Therefore, the question of determining OMPS-NM geolocation accuracy is to find the best correlated one with its original OMPS-NM observations among m $\times$ n (31 $\times$ 27) VIIRS-collocated OMPS-NM radiances. As shown in Figure 5, the spread or standard deviation between the VIIRS-collocated and real OMMS-NM radiances is very sensitive to the misalignment between OMPS-NM and VIIRS. Therefore, the correlation coefficients can be calculated from the m $\times$ n (31 $\times$ 27) datasets VIIRS-collocated OMPS radiance with the real OMPS radiance data individually, resulting in m $\times$ n (31 $\times$ 27) correlation coefficients. For each macropixel FOV position (from 0 to 34), a contour map can be generated in the along-track and cross-track perturbation angles of $\theta_{guess}^{fov}$ and $\varphi_{guess}^{fov}$. The left panel of Figure 9 shows an example of a correlation contour map at macropixel FOV position 0 using the granule in Figure 1a. The correlation coefficients between VIIRS-collocated and real OMPS-NM radiance vary with the along-track and cross-track perturbation angles, but one can find the maximum correlation of 0.99897 at the along-track and cross-track perturbation angles of $(4.9°, -0.2°)$. Clearly demonstrated in Figure 8 is a scatter plot between the VIIRS-collocated and real OMPS-NM radiances with and without perturbation. Particularly, based on the original OMPS-NM LOS vectors without any perturbation to collocate with VIIRS, the correlation coefficient between VIIRS and OMPS-NM radiances is 0.91411 and there is a large spread (indicated by the blue dots in Figure 8). However, when the original LOS vectors were perturbed by a pair of angles $(4.9°, -0.2°)$, their correlation coefficient reaches

to the maximum (shown by the red dots in Figure 8). In other words, one can state that, compared to the VIIRS geolocation dataset, the OMPS-NM LOS vector should be off 4.9° in the along-track direction and −0.2° in the cross-track direction for the scan position 0 in the S/C coordinate to match VIIRS. A similar practice can be carried out for all macropixel FOV positions, as shown in Figure 10 as an example.

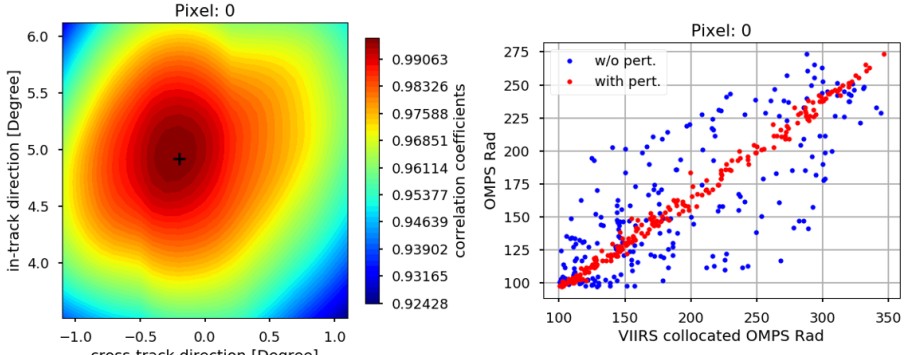

**Figure 9.** Contour plots (**left**) of the correlation coefficients between VIIRS-collocated and real OMPS-NM radiances varying with the perturbation angles in the along-track and cross-track directions for macropixel FOV 0, where the plus marker shows the maximum correlation coefficient. A scatter plot (**right**) between VIIRS-collocated and real OMPS-NM radiances with and without LOS vector perturbation.

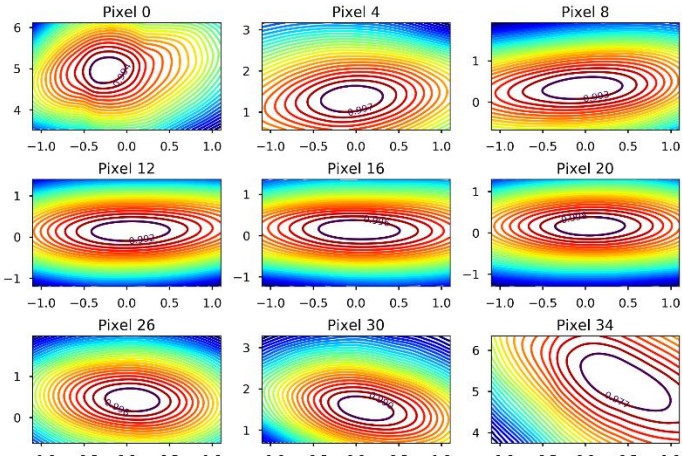

**Figure 10.** Contour plots of the correlation coefficients between VIIRS-collocated and real OMPS-NM radiances varying with the perturbation angles in the along-track and cross-track directions at the nine (9) macropixel FOV positions, where the angle values with the maximum correlation are used to estimate the geolocation error of OMPS-NM.

Figure 11 shows the geolocation assessment results from the data from 29 May 2021 (shown in Figure 1). Note that the dash lines give the ±1/2 macropixel FOV size in the cross-track and along-track resolutions. It clearly shows that the OMPS-NM geolocation errors are mostly caused by the off-nadir pixels in the along-track direction, which is off around 4 pixels (~5.0°) at FOV 0 and FOV 34. This error quickly reduces from the off-nadir position (4.9° at FOV 0 and 5.3° at FOV 34) to the nadir position (close to 0 degrees). In the cross-track track direction, except for the beginning and end of FOVs (FOVs 0–4 and 31–34), which are off 5–15% FOV size, the offset angles for other FOVs are along the zero line.

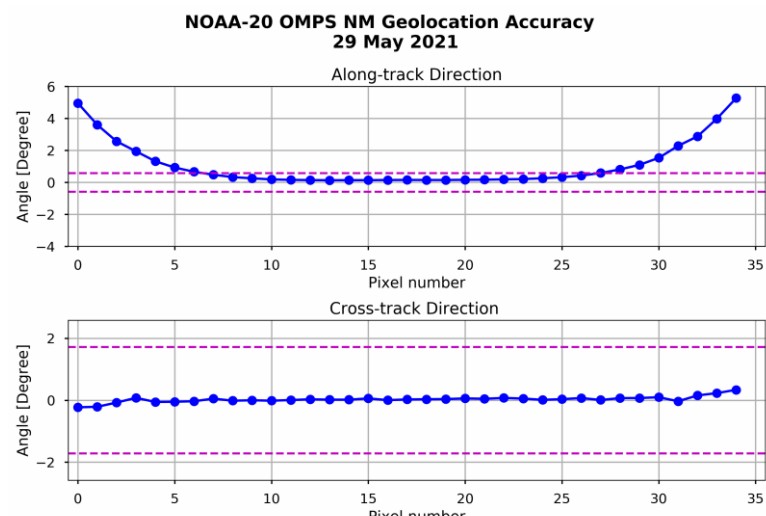

**Figure 11.** Geolocation assessment results, including the offset angles in the along-track and cross-track directions, where the dashed lines indicate the ±1/2 macropixel FOV size in terms of degrees.

The flowchart in Figure 12 summarizes the procedures for quantifying OMPS-NM geolocation accuracy using VIIRS M1 band geolocation fields. Both VIIRS and OMPS-NM geolocation fields on the Earth's surface are transformed back to the S/C coordinate through inverse geolocation computation, respectively. The closed cycles indicate the perturbation procedures that generate m × n OMPS-NM LOS vectors (including FOV shapes) in the S/C coordinate. The collocation between VIIRS and OMPS-NM is carried out in the S/C coordinate. The best matched vector among the m × n (31 × 26) OMPS-NM LOS vectors is selected by examining the maximum correlation between VIIRS-collocated and real OMPS-NM radiance. Based on this best-matched vector, the offset angle in the along-track and cross-track can be equally derived to characterize the OMPS-NM geolocation accuracy at each macropixel FOV position.

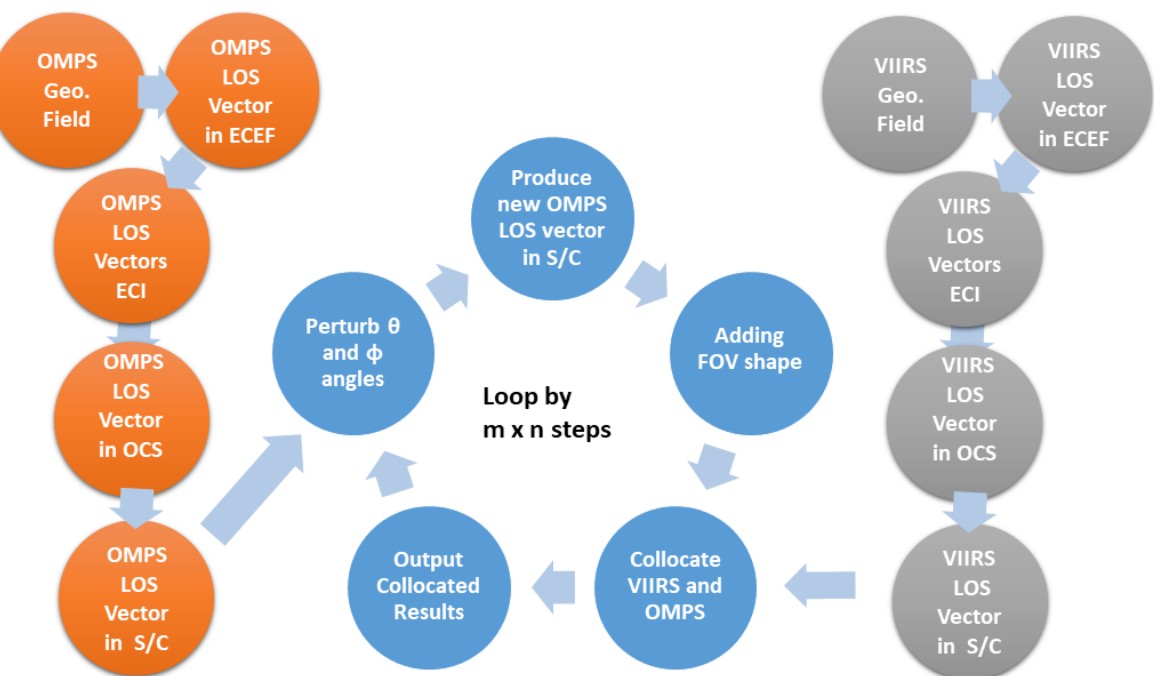

**Figure 12.** Flowchart describing the method of quantifying OMPS-NM geolocation accuracy using VIIRS M1 band geolocation data.

## 5. Application

The application of the geolocation assessment method to three datasets with different spatial configurations (Table 1) is presented in this section. In addition, the method to optimize the view angle lookup table based on the assessment results is discussed below.

### 5.1. Application for SNPP and NOAA-20 Operational OMPS-NM Data

The method presented in the above section is used to quantify the geolocation accuracy of the NOAA operational OMPS SDR products for SNPP and NOAA-20. It should be noted that SNPP OMPS-NM has a lower spatial resolution (50 × 50 km) compared to NOAA-20 (50 × 17 km). The data span from 27 May 2021 to 26 March 2022, when the manuscript was finalized. To further test the robustness of the method, we randomly chosen granules located in different geographic regions (middle and low latitudes) with different conditions (over land or ocean) every day. Each case is composed of 150–200 OMPS-NM lines with non-uniform scenes (e.g., cloudy scenes over the ocean) to cover the relatively large dynamic range. Figures 13 and 14, respectively, show the time series of geolocation assessment results for NOAA-20 and SNPP OMPS-NM at different FOVs (FOV 0—at the beginning of the scan, FOV 17—at nadir, and FOV 34—at the end of scan). The dashed lines in the figures give the OMPS-NM FOV size in the cross-track and along-track resolutions. The time series clearly shows that, for both SNPP and NOAA-20 OMPS-NM, the geolocation errors are mostly from the off-nadir pixels in the along-track direction, which is off ~5.0 degrees (around 4 FOV size). The errors were greatly reduced after 28 June 2021 due to the fact that the view angle lookup table was updated. On the other hand, the geolocation errors in the cross-track direction are well below the FOV size and are consistent with time. Figure 15 shows the new geolocation assessment results on 24 March 2022, after the view angle table was updated. The scan-dependent geolocation errors in the along-track direction are greatly reduced. This clearly shows that the geolocation accuracy is on the sub-pixel level in both cross-track and along-track directions. The worst performance is at the end of scan position (pixel 34) in the along-track direction, which is less than ¼ pixel size. On the other hand, a yaw pattern remains in the along-track direction, indicating that there is still room to further adjust the view angles if needed.

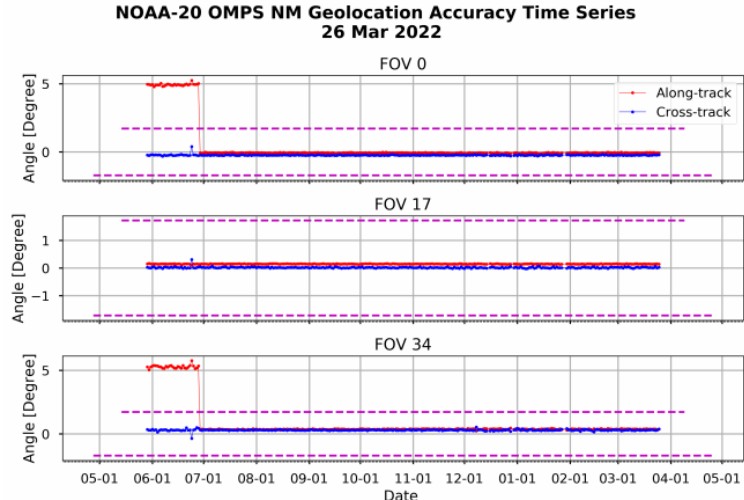

**Figure 13.** Time series of the assessment results for NOAA-20 OMPS-NM at macropixel FOV 0, 17 and 25, where different *y*-axis scales are used for three plots. The dashed lines indicate the ±1/2 FOV size in terms of degree. The time series extended from 27 May 2021 to 26 March 2022 (309 days) and the data in each day contains 150–200 OMPS-NM lines with randomly selected non-uniform scenes.

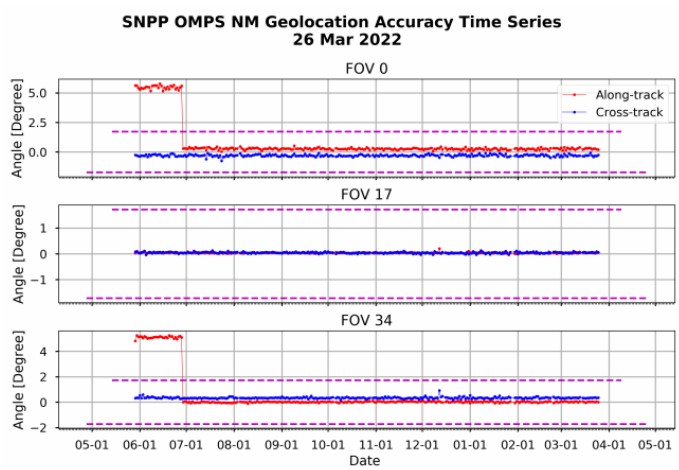

**Figure 14.** Same as Figure 13 but for SNPP OMPS-NM.

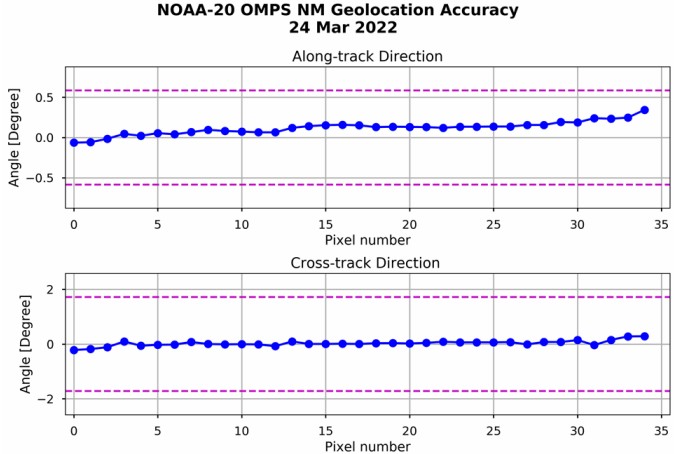

**Figure 15.** Same as Figure 11 but on 24 March 2022 after the view angle table was updated.

Currently, the above method has been fully operational under the NOAA Integrated Calibration and Validation System (ICVS) (https://www.star.nesdis.noaa.gov/icvs-beta/status_N20_OMPS_NM.php) (accessed on 1 June 2021). Therefore, it can near real-time track and monitor the geolocation accuracy of OMPS-NM daily and its variation with time. Finally, the proposed method can make OMPS-NM and VIIRS better align, thus facilitating an application that relies on the combination of OMPS-NM and VIIRS measurements and products.

### 5.2. Potential Application for Future High-Resolution OMPS-NM Data

As discussed in the introduction section, the spatial resolution of the future OMPS-NM on JPSS-2 will be further improved in both cross-track (with less CCD pixels aggerated) and along-track (with less integrating time) directions. Therefore, it is important to test whether the proposed method can work for future high-resolution OMPS-NM data. Therefore, the medium-resolution OMPS-NM data on NOAA-20—which were generated from NASA's Science Investigator-led Processing System (SIPS) and has a spatial resolution of 17 km (along-track) by 12.5 km (cross-track) at nadir—is used as a proxy to test the proposed method.

Using a randomly selected granule on 15 August 2021 as an example, Figure 16 shows the assessment procedures following the above method. Figure 16d, the radiance difference image between real and VIIRS-collocated OMPS-NM radiances based on the original OMPS geolocation fields, clearly shows that OMPS-NM is not perfectly aligned with VIIRS because of the noise patterns. However, as shown in Figure 16e, these noise features disappear after moving the OMPS-NM image around, making these two images perfectly aligned.

This improvement can be further demonstrated in Figure 16f—their radiance scatter plot before and after perturbation. Figure 17 shows the perturbation angles in the cross-track and along-track directions, which are basically the geolocation accuracy for the NASA medium-resolution OMPS-NM data. It is less than or close to ½ macropixel FOV size for all macropixel positions in both cross-track and along-track directions. It further demonstrates that the method can identify the geolocation error at the sub-pixel level even though the spatial resolution is greatly improved.

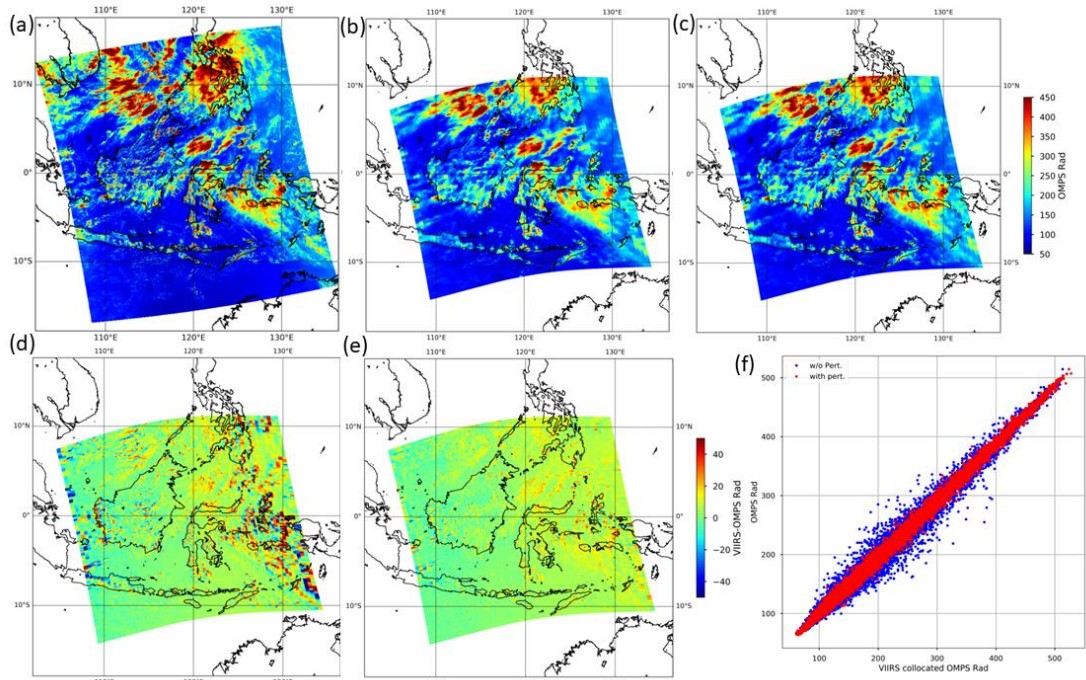

**Figure 16.** (**a**) VIIRS image at M01 band (412 nm) on 15 August 2021, (**b**) the correspondent medium resolution data on NOAA-20 image at 420 nm, (**c**) VIIRS-collocated image based on the OMPS-NM geometry characteristics, the radiance difference image between VIIRS-collocated and real OMPS-NM radiances, (**d**) based on the original OMPS geolocation fields and (**e**) with the ones with perturbation that perfectly are aligned with VIIRS, and (**f**) Scatter plot between real and VIIRS-collocated OMPS-NM radiances. In (**f**), the blue dots indicate the VIIRS-collocated OMPS-NM radiances using the original geolocation fields, while the red dots indicate those with perturbation. The radiance unit is $W/(cm^3 \, Sr)$.

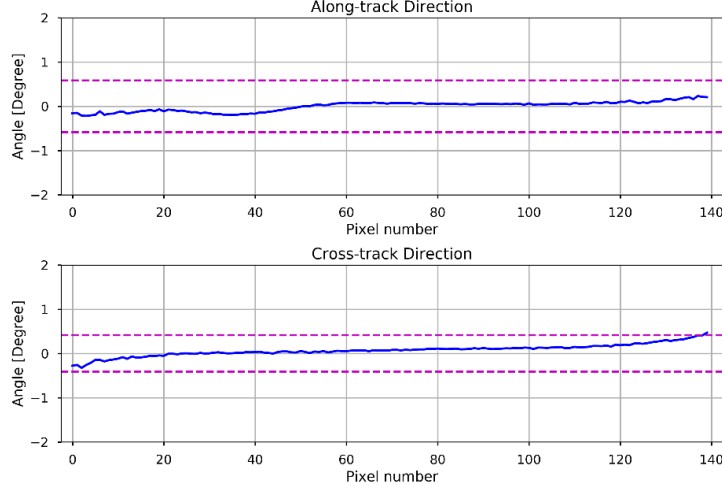

**Figure 17.** Same as Figure 11 but for the medium resolution OMPS-NM data on NOAA-20.

*5.3. View Angle Optimization*

In addition, the assessment results can be used to optimize the OMPS-NM field angle lookup tables to refine geolocation accuracy. Since these field angles were measured during the prelaunch test, it is possible that these angles can be changed after launch, such as launch shift, gravity effects, and thermal distortion. Thus, it is significant to perform postlaunch geometric calibration to refine calibration parameters (similar to postlaunch radiometric and spectral calibrations). As we discussed above, there are two sets of parameters used in the sensor-level geolocation computations, i.e., the optical view angle lookup table and the instrument mounting matrix, both of which are static. As we point out in the above section, the transformation of the mounting matrix is very small (on the order of $10^{-3}$ degree), which is not likely to cause the large geolocation errors revealed from the assessment results. Therefore, the focus is on field view angles.

As shown in Figure 11, along each macropixel FOV position, we can identify the two offset angles of ($\theta$, $\varphi$) and thus compute the true LOS unit vector [$x$, $y$, $z$] in the S/C coordinate following Equation (4). By multiplying by the mounting matrix, one can further transform the unit vector LOS [$x$, $y$, $x$] from the S/C coordinate into the instrument coordinate as [$x'$, $y'$, $x'$]. Using Equation (1), given the existing view vector [$x'$, $y'$, $z'$], the two view angles $\alpha$ and $\beta$ at each position can be calculated. The derived angles can replace those from the prelaunch test and further improve its geolocation performance. A similar practice was carried out for both SNPP and N20 OMPS-NM and their view angles were updated on 28 June 2021.

## 6. Conclusions

The data quality of OMPS-NM SDR products is essential for their downstream geophysical parameter retrievals. Extending the method by Wang et al. [12,13], this study presents a method for OMPS-NM geolocation assessment using high spatial resolution VIIRS band M1 data through image registration. The basic idea is to find the best alignment position with maximum correlation between VIIRS and OMPS-NM radiances by perturbing OMPS-NM LOS vectors in the cross-track and along-track directions with small steps in the S/C coordinate. The offset angles at the best collocation position that perfectly align OMPS-NM and VIIRS are then used to characterize OMPS-NM geolocation accuracy. To demonstrate the methodology, the proposed method is successfully employed to evaluate the OMPS-NM geolocation accuracy with different spatial resolutions for both NOAA-20 and SNPP.

Presently, the new method has been fully operational under the NOAA ICVS. Based on the long-term assessment results over almost one year, the following findings are disclosed:

(1) The geolocation errors at the off-nadir positions are substantially reduced after the view angle table was updated on 28 June 2021.
(2) With the updated view angle table, the geolocation accuracy for both SNPP and NOAA-20 OMPS-NM is on the sub-pixel level in both cross-track and along-track directions along all the scan positions, which is less than ¼ pixel (the worst scenario).
(3) The geolocation accuracy is stable, varying with time with an updated view angle table.

The method proposed in this study places the framework in place to assess the geolocation accuracy of future high-resolution OMPS-NM measurements. Finally, the new method can perform postlaunch geometric calibration by optimizing the CCD field view angles based on the assessment results. Therefore, the new method is expected to play a critical role in future JPSS OMPS NM geolocation accuracy assessments.

**Author Contributions:** Conceptualization, L.W., C.P. and B.Y.; methodology, L.W. and C.P.; software, L.W.; validation, L.W. and C.P.; formal analysis, L.W.; investigation, L.W. and C.P.; data curation, L.W., T.B. and J.C.; writing—original draft preparation, L.W.; writing—review and editing, L.W., C.P., B.Y. and S.K.; visualization, L.W.; supervision, B.Y.; project administration, B.Y. and L.Z.; funding acquisition, L.Z., S.K. and M.G. All authors have read and agreed to the published version of the manuscript.

**Funding:** This study was supported by NOAA grant NA19NES4320002 (Cooperative Institute for Satellite Earth System Studies—CISESS) at the University of Maryland/ESSIC by the NOAA JPSS program office.

**Data Availability Statement:** The NOAA OMPS-NM and VIIRS data used in this study were downloaded from the NOAA's Comprehensive Large Array Data Stewardship System at https://www.class.noaa.gov. The medium-resolution data of NOAA-20 OMPS-NM were obtained from the NASA's Science Investigator-led Processing System (SIPS) system at https://omisips1.omisips.eosdis.nasa.gov (with request). The forward and inverse geolocation computation software is available at https://github.com/wanglikun1973. (all accessed on 1 October 2020).

**Acknowledgments:** The authors thank Glen Jaross and Seftor Colin from NASA/GFSC for sharing their presentation about discovering the OMPS NM off-nadir geolocation errors and correction analyses, Lawrence Flynn from NOAA/NESDIS/STAR for his critical comments and suggestions, and Ninghai Sun and Ding Liang for their help when the first author implemented the geolocation assessment software under ICVS.

**Conflicts of Interest:** The manuscript contents are solely the opinions of the authors and do not constitute a statement of policy, decision or position on behalf of the NOAA or the U.S. government. The authors declare no conflict of interest.

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
