# Peer review of "Geolocation Assessment and Optimization for OMPS Nadir Mapper: Methodology"

_remotesensing, doi:10.3390/rs14133040_

Round 1
Reviewer 1 Report
Grammar needs improving.
The structure needs improving. What are the aims? What have others done in the field? What are the methods/results/conclusions? Introduction inadequate. Too many self-citations. Instruments and datasets, irrelevant detail & repetition. 3 Nature of geolocation should be brought into the introduction, methods are inadequate and repetitive. Results included with methods.
Reviewer 2 Report
Very nice manucscipt. The authors have presented a method that can improve the OMPS geolocation accuracy by correlated the OMPS image and the VIIRS image. The authors deduce the parameters in the S/C frame by transforming the mis-alighments betweent the viirs and omps images. And then re-calculate the geolcation parameters. And the paper shows great improvements and are very useful for the satellite data users and researchers.
Author Response
We thank this reviewer for his comments. There is no specific comments to address.
Reviewer 3 Report
The paper is well written and structured. It presents a method that, in a similar form, has already been applied to data from other instruments (CrIS). The method is applied to data from a different instrument in this paper and the fact that it is used in operational processing and resulted already in an update of calibration tables clearly justifies a publication. There are only a few minor points that could/should be corrected or improved:
Line 21/22: [..] where OMPS-NM and 21 VIIRS are optimally aligned are used [..]
Line 169: [..] where view angles [..]
Equations (1), (3), (4): the formatting of the numbering is off, the numbers are way to close to the equation itself
Equation (2): should be arctan instead of acrtan
Line 309: [..] the retrieved along track [..]
Line 432: [..] noted that SNPP OMPS-NM has a lower [..]
Figures 13 and 14: Maybe you can provide the number of datasets used for the statistics (at least order of magnitude) in the description
Section 5.3: As there is a section about the update of the view vectors, I would expect you to mention in this section, that and when the LUTs have been updated.
You are not consistent in naming the FOVs: sometimes it is micropixel (lines 237, 249, 255, 305), sometimes it is macropixel (lines 57, 105, 109, 158, 270, 282, 488, table 1)
Round 2
Reviewer 1 Report
The authors have not addressed the previous comments.
This manuscript is a resubmission of an earlier submission. The following is a list of the peer review reports and author responses from that submission.